# Genetic Correlation of miRNA Polymorphisms and STAT3 Signaling Pathway with Recurrent Implantation Failure in the Korean Population

**DOI:** 10.3390/ijms242316794

**Published:** 2023-11-27

**Authors:** Jung Hun Lee, Eun Hee Ahn, Min Jung Kwon, Chang Su Ryu, Yong Hyun Ha, Eun Ju Ko, Jeong Yong Lee, Ji Young Hwang, Ji Hyang Kim, Young Ran Kim, Nam Keun Kim

**Affiliations:** 1Department of Biomedical Science, College of Life Science, CHA University, Seongnam 13488, Republic of Korea; wjdgns960@gmail.com (J.H.L.); 0906sally@naver.com (M.J.K.); regis2040@nate.com (C.S.R.); hayo119@naver.com (Y.H.H.); ejko05@naver.com (E.J.K.); smilee3625@naver.com (J.Y.L.); 2Department of Obstetrics and Gynecology, CHA Bundang Medical Center, CHA University, Seongnam 13496, Republic of Korea; bestob@chamc.co.kr (E.H.A.); bin0902@chamc.co.kr (J.H.K.); 3Department of Genetics, Development, and Cell Biology, Iowa State University, Ames, IA 50011, USA; 4Department of Obstetrics and Gynecology, Fertility Center of CHA Gangnam Medical Center, CHA University, Seoul 06135, Republic of Korea; jyhwang@chamc.co.kr

**Keywords:** single nucleotide polymorphism, recurrent implantation failure, STAT3, microRNA, in vitro fertilization

## Abstract

The growing prevalence of in vitro fertilization-embryo transfer procedures has resulted in an increased incidence of recurrent implantation failure (RIF), necessitating focused research in this area. STAT3, a key factor in maternal endometrial remodeling and stromal proliferation, is crucial for successful embryo implantation. While the relationship between STAT3 and RIF has been studied, the impact of single nucleotide polymorphisms (SNPs) in miRNAs, well-characterized gene expression modulators, on STAT3 in RIF cases remains uncharacterized. Here, we investigated 161 RIF patients and 268 healthy control subjects in the Korean population, analyzing the statistical association between miRNA genetic variants and RIF risk. We aimed to determine whether SNPs in specific miRNAs, namely miR-218-2 rs11134527 G>A, miR-34a rs2666433 G>A, miR-34a rs6577555 C>A, and miR-130a rs731384 G>A, were significantly associated with RIF risk. We identified a significant association between miR-34a rs6577555 C>A and RIF prevalence (implantation failure [IF] ≥ 2: adjusted odds ratio [AOR] = 2.264, 95% CI = 1.007–5.092, *p* = 0.048). These findings suggest that miR-34a rs6577555 C>A may contribute to an increased susceptibility to RIF. However, further investigations are necessary to elucidate the precise mechanisms underlying the role of miR-34a rs6577555 C>A in RIF. This study sheds light on the genetic and molecular factors underlying RIF, offering new avenues for research and potential advancements in the diagnosis and treatment of this complex condition.

## 1. Introduction

In Korea, there has been a substantial increase in the number of patients seeking infertility treatment support. This increase can be attributed to the government’s enhanced support in covering infertility treatment expenses for couples facing reproductive challenges [1]. As the number of patients undergoing in vitro fertilization-embryo transfer (IVF-ET) continues to grow, the rising incidence of recurrent implantation failure (RIF) underscores the challenges of achieving successful implantation, despite the rapid advancements in assisted reproductive technology in recent years. Therefore, to address the limitations of this rapidly evolving technology, further research on RIF is essential.

The diagnosis of RIF is established when a patient fails to achieve a clinical pregnancy after undergoing three cycles of IVF-ET, which includes the transfer of four high-quality embryos to the endometrium [2]. To identify the most probable causes of RIF, various risk factors have been considered, including immunological factors, thrombophilias, anatomical abnormalities, and embryo aneuploidy [3]. Nonetheless, the precise etiology of RIF remains elusive, imposing limitations not only on the study of the condition itself but also on the application and interpretation of novel discoveries related to this disorder. Among the various stages of pregnancy, implantation emerges as one of the most critical phases.

Successful embryo implantation relies on two critical factors: a receptive uterus and an implantation-competent blastocyst [4]. During implantation, the presence of inflammatory gradients facilitates the apposition of the blastocyst to the uterine endometrium, followed by its adhesion to the surface epithelium of the endometrium [5,6]. Our objective was to establish a connection between factors regulating the expression of inflammation-associated genes and the prevalence of RIF. Therefore, we focused on the role of signal transducer and activator of transcription 3 (STAT3), a gene associated with inflammation. STAT3 is expressed in both uterine epithelial and stromal cells and plays a vital role in stromal proliferation, stromal differentiation, and uterine epithelial junctional reorganization [7]. These processes are essential for creating a receptive uterine environment for implantation.

STAT3, a member of the STAT family, plays a crucial role as a transcription factor transmitting signals from pro-inflammatory cytokines of the IL-6 family, such as IL-6, IL-11, and leukemia inhibitory factor (LIF) [8]. Within the uterine environment, the phosphorylation and activation of STAT3 are essential for embryo implantation. A previous study unveiled the distinct pathways through which STAT3 influences uterine receptivity and embryo attachment in both the uterine epithelium and stroma. Epithelial STAT3 is responsible for controlling the formation of a slit-like structure in the uterine lumen. In contrast, stromal STAT3 suppresses estrogen responsiveness and regulates epithelial proliferation [4,7]. Understanding the regulation of STAT3 gene expression by various factors will clarify its association with RIF. Therefore, we scrutinized the factors that could modulate STAT3 gene expression, such as microRNAs (miRNAs).

miRNAs are highly conserved non-coding RNA molecules, typically ranging in size from 19 to 25 nucleotides, which play a crucial role in regulating gene expression [9]. This regulation primarily involves negative control through mRNA cleavage, deadenylation, or the inhibition of translational repression. Positive control through the targeting of gene promoters is a rare occurrence [10]. miRNAs are indispensable for normal cellular functions as they bind to the 3′-untranslated regions (UTRs) of numerous target genes, thereby regulating a wide spectrum of genes. These regulatory characteristics highlight the deep involvement of miRNAs in most biological processes and their association with a wide range of pathological processes, spanning from myocardial infarction to autoimmune diseases [11].

Recent studies indicate that miRNAs in the endometrium play significant roles in the early stages of placenta formation and gestation, thus influencing implantation [12]. This underscores the potential of these miRNAs to serve as crucial elements for detecting implantation disorders and investigating the underlying causes of unexplained female infertility.

In the context of this study, our focus was on miRNAs targeting genes within the STAT3 pathway, to evaluate their capacity to influence gene expression. Specifically, we selected miRNAs that bind to the 3′-UTR of genes within the STAT3 pathway such as IL-6R and STAT3, assessing their potential impact on gene expression alteration. Among these miRNAs that influence STAT3 expression, we chose four specific miRNA SNPs: miR-218-2 rs11134527 G>A, miR-34a rs2666433 G>A, miR-34a rs6577555 C>A, and miR-130a rs731384 G>A. These miRNAs were of particular interest due to their potential connections with implantation [13]. Subsequently, we assessed the frequencies of these miRNA SNPs in RIF patients and control subjects within the Korean population.

While these four miRNA SNPs have been previously examined for their association with other diseases, their specific relationship with implantation and pregnancy remains undetermined [14,15,16]. Investigating the link between miRNA SNPs and RIF occurrence represents an innovative approach to identifying potential biomarkers for RIF diagnosis and therapeutic targeting. To the best of our knowledge, this study marks the first attempt to uncover the impact of these selected miRNA SNPs on RIF prevalence in Korean women.

## 2. Results

### 2.1. Clinical Profiles of Study Subjects

We examined the clinical characteristics of both control and RIF patient groups to establish a foundational understanding of the study population. Table 1 presents the clinical profiles of 268 control subjects and 161 RIF patients. Following age frequency matching, there were no statistically significant differences in age distribution between the control and patient groups (*p* = 0.191; Table 1). However, significant differences were observed in prothrombin levels, blood urea nitrogen, creatinine, estradiol, TSH, and LH levels between the patient and control groups, all of which were significantly higher in the RIF patient group compared to the control group (*p* ≤ 0.0003; Table 1). Importantly, no significant differences were observed in any other clinical characteristic between the groups.

### 2.2. Comparison of Genotype Frequencies of miRNA SNPs

We aimed to determine the relationship between specific miRNA SNPs and the risk of RIF, offering crucial genetic insights into the condition. Table 2 displays the genotype frequencies of the miRNA SNPs miR-218-2 rs11134527 G>A, miR-34a rs2666433 G>A, miR-34a rs6577555 C>A, and miR-130a rs731384 G>A in both control subjects and RIF patients. Age-adjusted ORs were obtained through logistic regression analysis. Importantly, all genotype frequencies for these SNPs in both control and patient groups adhered to the Hardy–Weinberg equilibrium (*p* > 0.05), indicating that the population distribution aligned with the Hardy–Weinberg equation.

Our analysis, which involved comparing genotype frequencies between control subjects and RIF patients and incorporating data from logistic regression, highlighted a significant association between the miR-34a rs6577555 C>A AA genotype and an increased disease prevalence (AA: IF ≥ 2: AOR = 2.264, 95% CI = 1.007–5.092, *p* = 0.048; Table 2). Furthermore, upon investigation of genotype frequencies based on the number of IFs among RIF patients, we found that patients with miR-34a rs6577555 C>A AA and recessive genotypes who experienced a higher number of IFs were at a greater risk of developing the disorder (AA: IF ≥ 3 group: AOR = 2.322, 95% CI = 1.052–5.125, *p* = 0.034; IF ≥ 4: AOR = 2.783, 95% CI = 1.185–6.536, *p* = 0.019; CC+CA vs. AA: IF ≥ 3 group: AOR = 2.406, 95% CI = 1.068–5.419, *p* = 0.037; IF ≥ 4: AOR = 2.457, 95% CI = 1.077–5.606, *p* = 0.033; Table 3). However, there were no statistically significant differences in the other miRNA SNPs (miR-218-2 rs11134527 G>A, miR-34a rs2666433 G>A, and miR-130a rs731384 G>A) between the control and patient groups. This analysis revealed a significant association between the miR-34a rs6577555 C>A AA genotype and an elevated risk of RIF, suggesting the potential genetic influence of this variant on RIF development.

### 2.3. Combination Analysis

We analyzed allele combinations for the four miRNA SNPs to explore potential interactions between polymorphic sites that might have synergistic effects on RIF risk (Table 3). Among the various allele combinations, specific haplotype combinations, such as A-A-C-A haplotype for miR-218 (rs11134527 G>A)/miR-34a (rs2666433 G>A)/miR-34a (rs6577555 C>A)/miR-130a (rs731384 G>A) and G-G-A haplotype for miR-218 (rs11134527 G>A)/miR-34a (rs2666433 G>A)/miR-130a (rs731384 G>A), exhibited statistical significance in relation to the risk of RIF (*p* < 0.05, Table 4). Moreover, the A-G-A haplotype of miR-218 (rs11134527 G>A)/miR-34a (rs2666433 G>A)/miR-34a (rs6577555 C>A) involving three polymorphic loci exhibited synergistic effects in increasing the risk of RIF (AOR = 1.561, 95% CI = 1.001–2.433, *p* = 0.049; Table 4).

In addition, a combined genotype analysis of miRNA SNPs was conducted (Table 4), revealing that the combination of miR-34a rs6577555 C>A AA and miR-130a rs731384 G>A GG genotypes significantly increased the risk of RIF. This specific genotype combination analysis, conducted for both RIF patients and control subjects, demonstrated the highest AOR with statistical significance regarding the risk of RIF (AA/GG: AOR = 2.881, 95% CI = 1.132–7.332, *p* = 0.026; Table 5). However, other genotype combinations did not yield statistically significant results in either combination analysis. Detailed results for all models, including allele and genotype combinations, are provided in Appendix A. Taken together, the A-G-A haplotype of miR-218 rs11134527 G>A/miR-34a rs2666433 G>A/miR-34a rs6577555 C>A and AA/GG genotype of miR-34a rs6577555 C>A/miR-130a rs731384 G>A exhibited synergistic effects to increase the risk of RIF, highlighting the importance of considering combinations of miRNA SNPs.

### 2.4. Variations in Clinical Parameters Based on miRNA Variants

RIF is a multifaceted clinical condition with diverse underlying factors that can impact its prevalence and prognosis. Given this complexity, several clinical parameters may play a role in the risk of RIF. To explore the potential influence of clinical parameters, we conducted an ANOVA to investigate the relationship between specific miRNA SNPs and various clinical factors associated with the risk of RIF. Table 6 highlights statistically significant differences in the variances across average levels of factors such as TSH, prolactin, CD8, and CD19, which are critical for successful implantation. These variances were observed among different genotypes of miR-34a rs6577555 C>A (*p* < 0.05). A comprehensive analysis, including all clinical factors, can be found in Appendix A.

Notably, RIF patients with the miR-34a rs6577555 C>A AA genotype exhibited relatively lower TSH levels compared to individuals with the CC and CA genotypes. Furthermore, significant differences in CD19 B-cell counts were observed among patient groups with distinct miR-34a rs6577555 C>A genotypes. The AA genotype was associated with an increased B-cell count, which may not favor successful implantation due to its potential involvement in creating adverse immune conditions during the pre-implantation phase of pregnancy, primarily through autoantibodies targeting nuclear components [17]. Thus, variations in clinical parameters, particularly TSH and B-cell counts, were associated with specific miRNA variants, suggesting a link between these genetic variations and clinical factors crucial for implantation.

### 2.5. Stratified Analysis of miRNA SNPs Based on Risk Factors in Patients and Control Subjects

To gain a clearer understanding of the risk factors in different genotype groups, we conducted a stratified analysis of miRNA SNPs in both RIF patients and control subjects, considering specific risk factors. These analyses revealed strong associations between certain genotypes and an increased risk of RIF within specific subgroups.

The miR-34a rs6577555 C>A combined GA+AA genotype was strongly associated with an elevated risk of RIF in patients with TSH levels ≤ 0.849 mU/L (AOR = 20.243, 95% CI = 2.836–144.5, *p* = 0.003; Table 7). Moreover, the miR-34a rs2666433 G>A GA+AA genotype showed an increased risk of RIF in patients with an aPTT level < 33 s (AOR = 3.590; 95% CI, 1.014–12.71; *p* = 0.048; Table 6). Additionally, the miR-130a rs731384 G>A GA+AA genotype was associated with a higher risk of RIF in patients with Hgb levels ≤ 11.2 g/dl (AOR = 4.787; 95% CI, 1.212–8.906; *p* = 0.026; Table 7). Thus, the stratified analysis revealed strong associations between specific miRNA genotypes and increased RIF risk in subgroups defined by clinical factors, highlighting the complex interplay between genetics and risk factors.

### 2.6. Synergistic Interaction between miRNA Polymorphisms and Clinical Parameters

To explore the multifaceted factors contributing to RIF, we examined the synergistic interplay between miRNA SNPs and clinical variables in combination (Appendix A). The establishment of cutoff values considered the influence of clinical value variations on RIF. For parameters such as PT and BUN levels, we identified the highest 15% of values (PT 11.7 s and BUN 12.6 mg/dl) in both the patient and control groups as threshold values. Our findings revealed a synergistic effect between miR-34a rs6577555 C>A and elevated PT and BUN levels, correlating with increased susceptibility to RIF (CA+AA with high PT level, AOR = 4.334, *p* = 0.008; CA+AA with high BUN level, AOR = 4.818, *p* = 0.015, Figure 1). Therefore, a combination of miRNA SNPs and clinical parameters may contribute to RIF susceptibility.

## 3. Discussion

Through previous studies examining the impact of various SNPs, it has become evident that genetic variants significantly influence the prevalence and prognosis of RIF [18,19,20,21]. Recent findings have also established connections between genetic variations and inflammation in susceptibility to RIF [22]. Inflammation is a vital component of the normal physiological process of implantation, and studies have suggested that an activated immune system plays a crucial role in facilitating successful implantation by promoting interactions between the embryo and uterine epithelium [5,23]. However, abnormal inflammatory responses can lead to reduced uterine receptivity and various pregnancy complications, including miscarriage, preterm birth, and IF [24]. Consequently, immune alterations represent a potential factor contributing to unexplained RIF.

Our previous investigations revealed that specific SNPs in genes associated with inflammation can substantially modify gene expression, thus profoundly affecting susceptibility to RIF [18]. Here, our primary focus was to examine the correlation between miRNA SNPs impacting STAT3 and the likelihood of developing RIF. This focus arises from STAT3’s role as a transcription factor that regulates the degree and duration of inflammation through the immunosuppressive cytokine interleukin 10 (IL-10) [25]. Given the anti-inflammatory role of STAT3 in the early implantation stage, we hypothesized that the miR-34a rs6577555 C>A SNP represents a functional SNP capable of controlling the necessary inflammatory responses required for successful early implantation.

Consistent with previous research regarding the role of STAT3 in the prevalence of RIF, our findings align with a growing body of evidence indicating that individuals with the miR-34a rs3577555 G>A AA genotype are at a higher risk for RIF compared to those with other genotypes [18]. This observation corroborates results from previous studies involving various cell types, where the introduction of miR-34a was shown to reduce STAT3 phosphorylation by directly targeting the mRNA of interleukin 6 receptor (IL-6R) [15,26]. The link between miR-34 and STAT3 suggests that the miR-34a AA genotype may exhibit an enhanced binding affinity to IL-6R, leading to decreased STAT3 gene expression. This alteration in miRNA function, particularly in individuals with the miR-34a rs3577555 G>A AA genotype, results in decreased STAT3 gene expression and triggers inflammation, ultimately increasing the risk of RIF.

In addition to genetic factors, several clinical parameters, such as TSH, PT, and BUN, play crucial roles in implantation and the maintenance of pregnancy. Notably, TSH levels exhibited a significant difference in miR-34a rs3577555 genotypes, particularly between the AA genotype and other genotypes. Our data suggest that individuals with the miR-34a rs3577555 G>A AA genotype tend to have significantly lower TSH levels; this observation was associated with an increased risk of RIF. Thyroid hormones, regulated by TSH, are critical for the proper development of the fetus and placenta during early pregnancy [27]. Thyroid disorders, including hypothyroidism and hyperthyroidism, have been linked to adverse pregnancy outcomes [28]. Specifically, a population-based study revealed that women with high thyroid hormone levels and low TSH levels are at a higher risk of spontaneous abortion or late pregnancy loss. Furthermore, infants born to these women have low birth weights and fetal thyrotoxicosis due to elevated maternal thyroid hormone levels [29]. Our study, along with existing research, suggests a complex interplay between decreased STAT3 expression and reduced TSH levels, both of which contribute to the risk of RIF [30]. Among the tested miRNA SNPs, miR-34a rs6577555 C>A, particularly in individuals with low TSH levels, shows significant potential as a predictive tool for RIF, as evidenced by the ROC analysis results (Appendix A).

Numerous studies provide compelling evidence supporting the utility of human peripheral blood miRNAs as significant diagnostic and prognostic biomarkers across a wide spectrum of diseases [31,32]. Consequently, the miRNAs examined in our study hold promise as biomarkers for the diagnosis and prognosis of RIF. Beyond diagnostic applications, miRNAs offer therapeutic potential, encompassing the use of miRNA mimics or anti-miRNAs to manipulate miRNA function. However, for the success of miRNA therapeutics, especially within the context of precision medicine, the development of patient-specific, low-toxicity, and stable miRNA delivery systems is of paramount importance [33].

This study bears certain limitations that warrant consideration for future research endeavors. First, the precise mechanisms through which these miRNA SNPs influence RIF development remain unclear. Gaining a comprehensive understanding of how these SNPs impact STAT3 regulation, whether through translational inhibition or mRNA degradation, requires further investigation. While we have suggested a potential synergistic effect between miRNA SNPs and TSH levels in increasing the risk of RIF, these findings are preliminary. The direct influence of miRNA SNPs and TSH levels on RIF pathophysiology remains ambiguous. Therefore, conducting additional in vitro and in vivo studies is imperative to validate their potential and achieve a more nuanced understanding to inform future applications. Second, to elucidate the associations revealed in this study, it is essential to account for potential confounding factors and additional environmental risk factors contributing to RIF. Third, our study population was limited to Korean women, thereby restricting the generalizability of our findings to a broader, more diverse global population. To establish broadly applicable results, further research must be conducted on a more diverse sample encompassing various ethnicities and geographical locations. Additionally, given the relatively low prevalence rate of RIF, our study was constrained by the availability of patient samples. Therefore, it is imperative to undertake research with a larger number of patient samples to reinforce and validate our findings.

## 4. Materials and Methods

### 4.1. Participant Selection and Exclusion Criteria

Blood samples were collected from RIF patients and control subjects from the Department of Obstetrics and Gynecology and the Fertility Center of CHA Bundang Medical Center (Seongnam, Republic of Korea) between March 2020 and December 2022. The study included a total of 161 RIF patients and 268 control subjects, all of Korean ethnicity. Ethical approval was granted by the Institutional Review Board of CHA Bundang Medical Center on 23 February 2010 (reference no. CHAMC2009-12-120), and written informed consent was obtained from all participants.

RIF was defined as the failure of implantation in more than two fresh IVF-ET cycles using embryos that had cleaved into more than 10 cells. Before embryo transfer, an embryologist examined all embryos, and only those of high quality were selected for transfer. IF was identified when the measured level of human chorionic gonadotropin (HCG) concentration was <5 U/mL at 14 days after the embryo transfer.

Patients diagnosed with IF due to maternal factors, such as anatomical, autoimmune, chromosomal, hormonal, infectious, and thrombotic causes, were excluded from the study. Exclusion criteria included uterine anatomical abnormalities (determined by computed tomography, hysterosalpingography, hysteroscopy, uterine sonography, or magnetic resonance imaging), autoimmune diseases (such as lupus and antiphospholipid syndrome; determined by examining lupus anticoagulants and anti-cardiolipin antibodies, respectively), chromosomal abnormalities (determined by karyotype analysis following standard protocols), hormone disorders (such as hyperprolactinemia, luteal insufficiency, and thyroid disease; determined by measuring levels of prolactin, thyroid-stimulating hormone (TSH), free T4, follicle-stimulating hormone (FSH), luteinizing hormone (LH), and progesterone in peripheral blood), and thrombophilia (determined by deficiencies in protein C and protein S and presence of anti-β2 glycoprotein antibodies).

Control subjects were selected based on the following criteria: regular menstrual cycles, normal karyotype (46XX), at least one naturally conceived pregnancy, and no history of pregnancy-related conditions, such as pregnancy loss or pre-eclampsia. The control subjects were recruited from the CHA Bundang Medical Center.

### 4.2. Assessment of Homocysteine, Folic Acid, Uric Acid, Blood Urea Nitrogen, Creatinine, and Blood Coagulation Status

Blood samples were collected from RIF patients following a 12-h fasting period to estimate the concentrations of biochemical factors. Homocysteine levels were determined using a fluorescence polarization immunoassay performed on an Abbott IMx analyzer (Abbott Laboratories, Abbott Park, IL, USA). Folic acid levels were measured through a competitive immunoassay using an ACS 180 Plus automated chemiluminescence system (Bayer Diagnostics, Tarrytown, NY, USA). Uric acid, blood urea nitrogen, and creatinine levels were measured using commercially available enzymatic colorimetric assays (Roche Diagnostics, GmbH, Mannheim, Germany). Platelet, white blood cell, and hemoglobin levels were measured using a Sysmex XE 2100 automated hematology system (Sysmex Corporation, Kobe, Japan). Prothrombin time (PT) and activated partial thromboplastin time (aPTT) were determined with an ACL TOP automated photo-optical coagulometer (LSI Medience, Tokyo, Japan).

### 4.3. Hormone Assays

Blood samples were obtained from subjects on the second or third day of their menstrual cycles through venipuncture. Serum samples were extracted from the collected blood samples before measurement. To measure estradiol, TSH, and prolactin levels, radioimmunoassays were conducted (Beckman Coulter, Fullerton, CA, USA). Enzyme immunoassays were employed to measure FSH and LH levels following the manufacturer’s instructions (Siemens, Munich, Germany).

### 4.4. Flow Cytometric Analysis of Immune Cells

Flow cytometric analysis of immune cells was conducted using a BD FACS Calibur flow cytometer and CellQuest software version 3.0 (BD Biosciences, Seoul, Republic of Korea). We utilized fluorescently labeled monoclonal antibodies (including fluorescein isothiocyanate (FITC), phycoerythrin (PE), peridinin chlorophyll protein (PerCP), and allophycocyanin (APC)) specific for CD3, CD4, CD8, CD19, CD16, and CD56 from BD Biosciences. Furthermore, anti-NKG2A-PE antibodies obtained from Beckman Coulter were used during the sorting process. Peripheral blood mononuclear cells (2.5 × 10^5^) were stained for cell-surface antigens at 4 °C in the dark for 30 min, followed by two washes in 2 mL of phosphate-buffered saline containing 1% bovine serum albumin and 0.01% sodium azide. Subsequently, the cells were fixed in 200 µL of a 1% formaldehyde solution (Sigma-Aldrich, St. Louis, MO, USA) before sorting.

### 4.5. SNP Selection

To identify miRNAs binding to overlapping regions of genes, within the STAT3 pathway, mRNA transcripts, we conducted an online search using the TargetScan (http://www.targetscan.org, accessed on 26 June 2023) and TarBase v.8 databases (https://dianalab.e-ce.uth.gr/html/diana/web/index.php?r=tarbasev8, accessed on 28 June 2023). The targeted mRNA sequences, primarily located in the 3′-UTR, were acquired from the National Center for Biotechnology Information (www.ncbi.nlm.nih.gov/, accessed on 26 June 2023). Our investigation revealed that miR-218 and miR-34a interact with the 3′-UTR of IL-6R mRNA, while miR-130a interacts with the 3′-UTR of STAT3 mRNA, resulting in the downregulation of STAT3 gene expression (Appendix A).

### 4.6. Genetic Analysis

Genomic DNA was extracted from a 3 mL anticoagulated peripheral blood sample obtained from the patient using the G-DEX blood extraction kit (iNtRON, Seongnam, Republic of Korea). Post-extraction, the genomic DNA underwent immediate assessment using NanoDrop-2000 (Thermo Fisher Scientific, Waltham, MA, USA) to determine both quantity (A260) and quality (A260/A280 ratio) of the isolated DNA. Additionally, the purity of genomic DNA was further validated through agarose gel electrophoresis to examine DNA fragment patterns.

SNP genotyping analysis for all miRNA SNPs was conducted using real-time polymerase chain reaction (PCR) and the TaqMan SNP Genotyping Assay Kit (Applied Biosystems, Waltham, MA, USA). In a concise procedure, real-time PCR was performed on the Rotor-Gene RG-3000 (Corbett Research, Sydney, Australia), following the manufacturer’s protocol. A mixture comprising 1 μL of genomic DNA (100 ng/μL), 7.5 μL TaqMan Genotyping Master Mix (Applied Biosystems), 0.75 μL TaqMan SNP Genotyping Assay (Applied Biosystems), and distilled water was prepared, reaching a final volume of 15 μL for real-time PCR. The experimental setup included appropriate negative controls. The thermal cycling conditions for this experiment were as follows: 95 °C for 10 min, followed by 40 cycles of denaturation at 95 °C for 15 s and annealing at 60 °C for 1 min [34,35].

To validate the genotyping data obtained through real-time PCR analysis, DNA sequencing was conducted on approximately 15% of the total samples. These randomly selected samples for each miRNA SNP were subjected to DNA sequencing using an ABI 3730XL DNA Analyzer (Applied Biosystems).

### 4.7. Statistical Analysis

Clinical subject characteristics were analyzed based on variable types. To compare two groups with normally distributed continuous data, we used an independent sample *t*-test and present the data as means ± standard deviations. Conversely, for categorical variables, differences were assessed using a chi-square test, and the data are presented as frequencies and percentages.

We examined the normality of continuous variables using the Kolmogorov–Smirnov test. If the *p*-value from the Kolmogorov–Smirnov test was <0.05, indicating a non-normal distribution, differences were evaluated with the Mann–Whitney test. Allele frequencies from the subjects were used to confirm the deviation of the sample from Hardy–Weinberg equilibrium. We compared genotype frequencies of each miRNA SNP between RIF patients and control subjects using logistic regression. The significance of the association between miRNA SNPs was assessed using odds ratios (ORs) and 95% confidence intervals (Cis). Adjustment of OR and 95% CI values was made by considering the subjects’ ages.

For different miRNA SNPs, we employed analysis of variance (ANOVA) and Kruskal–Wallis tests to compare variances across the means of various clinical values (platelets, PT, aPTT, homocysteine, folate, natural killer cells, and uric acid) between RIF patients and control subjects. Additionally, we conducted receiver operating characteristic curve (ROC) analyses to assess the association between miRNA SNPs and disease presence, with subgroup analyses that considered various environmental factors. The area under the curve (AUC) reflected the discriminatory ability of the diagnostic test. An AUC of 1.0 indicates 100% sensitivity and 100% specificity, while an AUC of 0.5 suggests a non-discriminative biomarker. Generally, an AUC value exceeding 0.75 signifies a significant biomarker with clinical potential.

Statistical analyses were performed using several statistical software tools, including GraphPad Prism 4.0 (GraphPad Software Inc., San Diego, CA, USA), HAPSTAT 3.0 (University of North Carolina, Chapel Hill, NC, USA), and MedCalc version 20.2 (MedCalc Software, Mariakerke, Belgium). The level of statistical significance was set at *p* < 0.05 for all statistical analyses conducted.

## 5. Conclusions

In summary, our study revealed a strong association between the miR-34a rs6577555 C>A SNP and the risk of RIF in Korean women, particularly concerning the number of IFs. Given STAT3’s involvement in both inflammation and implantation [18], we conducted a comprehensive analysis to examine the intricate relationship between miRNA SNPs (miR-218-2 rs11134527 G>A, miR-34a rs2666433 G>A, miR-34a rs6577555 C>A, and miR-130a rs731384 G>A) and their impact on STAT3 gene expression, thereby influencing patient susceptibility to RIF.

Our statistical analysis revealed a significant increase in the prevalence of the AA genotype of miR-34a rs6577555 C>A among RIF patients. Furthermore, as the number of IFs increased, so did the risk of RIF, solidifying the link between this SNP and RIF susceptibility. Our study introduces a unique approach, delving into the intricate relationships between specific miRNA SNPs and STAT3, which has shed light on their profound implications for RIF. This distinctive focus advances our understanding of the genetic and molecular factors that underlie RIF, opening new avenues for further research and potentially revolutionizing diagnostic and therapeutic strategies in the field. The substantial AOR value and elevated frequency strongly suggest a close association between the AA genotype of miR-34a rs6577555 C>A and an increased risk of RIF. Furthermore, our investigation uncovered synergistic interactions between the miR-34a rs6577555 C>A SNP and specific clinical factors. Certain clinical parameters, which have the potential to elevate the risk of RIF, hold promise as clinical biomarkers for tailoring personalized treatments in RIF management.

In conclusion, our findings have far-reaching significance, offering valuable insights into the diagnosis, treatment, and personalized care of RIF. This research represents a critical step forward in unraveling the complex interplay between genetics and clinical factors in the context of RIF, promising to benefit both patients and healthcare providers as we strive to enhance reproductive outcomes for women facing this challenging condition.

## Figures and Tables

**Figure 1 ijms-24-16794-f001:**
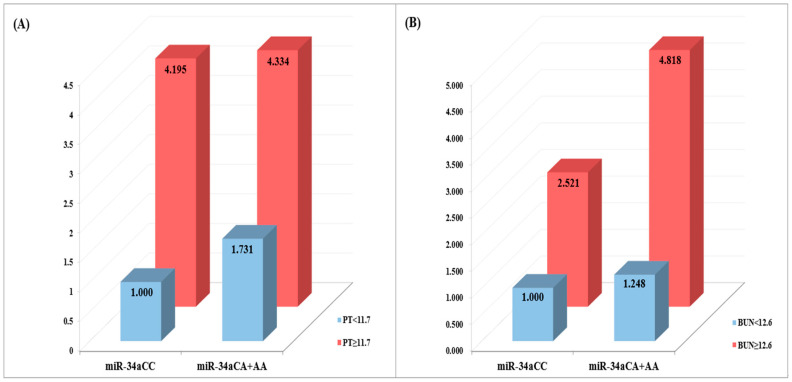
Synergistic effect of miR-34a rs6577555 variant with clinical parameters. (**A**,**B**) The panels show AOR of miR-34a rs6577555 with clinical parameters including prothrombin time (PT) (**A**), and blood urea nitrogen (BUN) (**B**).

**Table 1 ijms-24-16794-t001:** Clinical characteristics of RIF patients and control individuals.

Characteristic	Controls (n = 268)	RIF Patients (n = 161)	*p* ^a^
Age (years)	34.47 ± 2.65	34.75 ± 3.25	0.191 ^b^
BMI (kg/m^2^)	21.81 ± 3.41	21.44 ± 3.21	0.402
Previous implantation failure (n)	0	5.22 ± 2.38	N/A
Live births (n)	1.23 ± 0.46	0	N/A
Mean gestational age (weeks)	40.77 ± 2.20	0	N/A
PT (s)	10.56 ± 0.84	10.86 ± 2.17	**<0.0001 ^b^**
aPTT (s)	29.09 ± 3.69	29.59 ± 3.41	0.224
PLT (10^3^/μL)	239.06 ± 62.59	246.47 ± 69.11	0.280 ^b^
Homocysteine (μmol/L)	6.58 ± 2.24	6.75 ± 1.77	0.750
Folate (mg/mL)	14.32 ± 8.64	14.78 ± 8.08	0.848
Uric acid (mg/dl)	3.92 ± 1.00	4.01 ± 0.98	0.550
BUN (mg/dl)	8.88 ± 2.70	10.51 ± 2.87	**<0.0001**
Creatinine (mg/dl)	0.64 ± 0.16	0.79 ± 0.10	**<0.0001 ^b^**
E2 (Basal) (pg/mL)	26.48 ± 14.75	60.41 ± 114.61	**<0.0001 ^b^**
TSH (mU/L)	1.60 ± 1.01	2.22 ± 1.44	**0.0003 ^b^**
FSH (mU/L)	8.30 ± 2.82	9.24 ± 4.90	0.425 ^b^
LH (mU/L)	3.31 ± 1.82	4.90 ± 2.42	**<0.0001 ^b^**
Prolactin (ng/mL)	N/A	13.68 ± 7.16	N/A
WBC (10^3^/μL)	6.88 ± 2.40	7.28 ± 2.84	0.485 ^b^
Hgb (g/dl)	12.56 ± 2.13	12.52 ± 1.43	0.094 ^b^
CD3 (pan T) (%)	N/A	67.39 ± 10.97	N/A
CD4 (helper T) (%)	N/A	35.66 ± 9.33	N/A
CD8 (suppressor) (%)	N/A	28.21 ± 8.02	N/A
CD19 (B-cell) (%)	N/A	11.70 ± 4.79	N/A
CD56 (NKcell) (%)	N/A	17.41 ± 9.04	N/A

Note: Data are presented as the mean ± standard deviation; RIF, recurrent implantation failure; BMI, body mass index; PT, prothrombin time; aPTT, activated partial thromboplastin time; PLT, platelet count; BUN, blood urea nitrogen; E2, estradiol; TSH, thyroid-stimulating hormone; FSH, follicle stimulating hormone; LH, luteinizing hormone; WBC, white blood cell; Hgb, hemoglobin; N/A, not applicable. ^a^ *p*-values were calculated by two-sided *t*-test for continuous variables and chi-square test for categorical variables. ^b^ Mann–Whitney test for continuous data was undertaken when F-test *p*-value for equal variances was lower than 0.05.

**Table 2 ijms-24-16794-t002:** Comparison of genotype frequencies and AOR values for polymorphisms between the RIF patients and control subjects.

Genotypes	Controls (n = 268)	IF ≥ 2 (n = 159)	AOR (95% CI) ^a^	*p* ^a^
miR-218-2 rs11134527 G>A				
AA	94 (35.1)	49 (30.8)	1.000 (reference)	
AG	134 (50.0)	85 (53.5)	1.22 (0.786–1.893)	0.376
GG	40 (14.9)	25 (15.7)	1.228 (0.666–2.266)	0.510
Dominant (AA vs. AG+GG)			1.22 (0.801–1.858)	0.354
Recessive (AA+AG vs. GG)			1.069 (0.62–1.841)	0.811
HWE *p*	0.488	0.235		
miR-34a rs2666433 G>A				
GG	153 (57.1)	87 (54.7)	1.000 (reference)	
GA	97 (36.2)	66 (41.5)	1.208 (0.802–1.82)	0.365
AA	18 (6.7)	6 (3.8)	0.617 (0.235–1.623)	0.328
Dominant (GG vs. GA+AA)			1.115 (0.75–1.658)	0.590
Recessive (GG+GA vs. AA)			0.555 (0.215–1.432)	0.223
HWE *p*	0.624	0.127		
miR-34a rs6577555 C>A				
CC	161 (60.1)	89 (56.0)	1.000 (reference)	
CA	95 (35.4)	55 (34.6)	1.042 (0.683–1.59)	0.848
AA	12 (4.5)	15 (9.4)	2.264 (1.007–5.092)	**0.048**
Dominant (CC vs. CA+AA)			1.164 (0.781–1.736)	0.457
Recessive (CC+CA vs. AA)			2.164 (0.983–4.767)	0.055
HWE *p*	0.669	0.141		
miR-130a rs731384 G>A				
GG	202 (75.4)	122 (76.7)	1.000 (reference)	
GA	63 (23.5)	34 (21.4)	0.905 (0.563–1.455)	0.679
AA	3 (1.1)	3 (1.9)	1.657 (0.329–8.348)	0.540
Dominant (GG vs. GA+AA)			0.94 (0.592–1.492)	0.791
Recessive (GG+GA vs. AA)			1.695 (0.338–8.511)	0.521
HWE *p*	0.432	0.727		

Note: IF, implantation failure; AOR, adjusted odds ratio; HWE, Hardy-Weinberg equilibrium; 95% CI, 95% confidence interval; N/A, not applicable; ^a^ Adjusted by age.

**Table 3 ijms-24-16794-t003:** Genotype frequencies for each polymorphism according to the number of IFs.

Genotypes	Controls (n = 268)	IF ≥ 3 (n = 147)	AOR (95% CI) ^a^	*p* ^a^	IF ≥ 4 (n = 117)	AOR (95% CI) ^a^	*p* ^a^
miR-218-2 rs11134527 G>A							
AA	94 (35.1)	44 (29.9)	1.000 (reference)		34 (29.1)	1.000 (reference)	
AG	134 (50.0)	78 (53.1)	1.253 (0.795–1.975)	0.331	61 (52.1)	1.287 (0.782–2.117)	0.322
GG	40 (14.9)	25 (17.0)	1.397 (0.749–2.607)	0.293	22 (18.8)	1.678 (0.856–3.289)	0.132
Dominant (AA vs. AG+GG)			1.281 (0.83–1.978)	0.264		1.36 (0.846–2.187)	0.204
Recessive (AA+AG vs. GG)			1.182 (0.684–2.044)	0.549		1.348 (0.758–2.398)	0.310
HWE *p*	0.488	0.337			0.560		
miR-34a rs2666433 G>A							
GG	153 (57.1)	81 (55.1)	1.000 (reference)		63 (53.9)	1.000 (reference)	
GA	97 (36.2)	60 (40.8)	1.187 (0.779–1.809)	0.426	52 (44.4)	1.342 (0.856–2.106)	0.200
AA	18 (6.7)	6 (4.1)	0.674 (0.255–1.781)	0.426	2 (1.7)	0.308 (0.068–1.385)	0.125
Dominant (GG vs. GA+AA)			1.106 (0.736–1.662)	0.628		1.183 (0.762–1.838)	0.454
Recessive (GG+GA vs. AA)			0.612 (0.237–1.582)	0.311		0.258 (0.059–1.135)	0.073
HWE *p*	0.624	0.209			0.017		
miR-34a rs6577555 C>A							
CC	161 (60.1)	81 (55.1)	1.000 (reference)		59 (50.4)	1.000 (reference)	
CA	95 (35.4)	51 (34.7)	1.054 (0.683–1.626)	0.814	45 (38.5)	1.268 (0.796–2.02)	0.318
AA	12 (4.5)	15 (10.2)	2.406 (1.068–5.419)	**0.034**	13 (11.1)	2.783 (1.185–6.536)	**0.019**
Dominant (CC vs. CA+AA)			1.195 (0.794–1.8)	0.393		1.418 (0.912–2.205)	0.121
Recessive (CC+CA vs. AA)			2.322 (1.052–5.125)	**0.037**		2.457 (1.077–5.606)	**0.033**
HWE *p*	0.669	0.112			0.330		
miR-130a rs731384 G>A							
GG	202 (75.4)	112 (76.2)	1.000 (reference)		90 (76.9)	1.000 (reference)	
GA	63 (23.5)	32 (21.8)	0.936 (0.575–1.521)	0.789	24 (20.5)	0.884 (0.518–1.51)	0.652
AA	3 (1.1)	3 (2.0)	1.814 (0.359–9.163)	0.471	3 (2.6)	2.283 (0.449–1.598)	0.320
Dominant (GG vs. GA+AA)			0.977 (0.609–1.566)	0.921		0.949 (0.567–1.589)	0.843
Recessive (GG+GA vs. AA)			1.842 (0.366–9.267)	0.459		2.344 (0.464–1.855)	0.303
HWE *p*	0.432	0.690			0.373		

Note: IF, implantation failure; AOR, adjusted odds ratio; HWE, Hardy–Weinberg equilibrium; 95% CI, 95% confidence interval; N/A, not applicable; ^a^ Adjusted by age.

**Table 4 ijms-24-16794-t004:** Allele combinations analysis for miRNA polymorphisms in RIF patients and controls.

Allele Combinations	Controls (2n = 536)	Cases (2n = 322)	OR (95% CI)	*p*
miR-218-2 rs11134527 G>A/miR-34a rs2666433 G>A/miR-34a rs6577555 C>A/miR-130a rs731384 G>A				
A-G-C-G	154 (28.7)	85 (26.7)	1.000 (reference)	
A-G-C-A	30 (5.6)	10 (3.1)	0.615 (0.286–1.319)	0.208
A-G-A-G	57 (10.6)	48 (15.1)	1.552 (0.972–2.478)	0.065
A-G-A-A	9 (1.7)	2 (0.6)	0.410 (0.086–1.941)	0.339
A-A-C-G	66 (12.3)	32 (10.1)	0.894 (0.542–1.473)	0.659
A-A-C-A	4 (0.7)	8 (2.5)	3.687 (1.078–12.61)	**0.034**
A-A-A-G	3 (0.6)	0 (0.0)	0.263 (0.013–5.149)	0.553
G-G-C-G	96 (17.9)	54 (17.0)	1.037 (0.676–1.59)	0.868
G-G-C-A	8 (1.5)	11 (3.5)	2.535 (0.981–6.55)	**0.048**
G-G-A-G	43 (8.0)	24 (7.5)	1.029 (0.584–1.813)	0.922
G-G-A-A	7 (1.3)	9 (2.8)	2.370 (0.852–6.596)	0.090
G-A-C-G	48 (9.0)	37 (11.6)	1.421 (0.857–2.355)	0.172
G-A-C-A	11 (2.1)	0 (0.0)	0.080 (0.005–1.374)	**0.018**
G-A-A-G	0 (0.0)	2 (0.6)	9.192 (0.436–193.8)	0.127
miR-218-2 rs11134527 G>A/miR-34a rs2666433 G>A/miR-34a rs6577555 C>A				
A-G-C	182 (34.0)	92 (28.9)	1.000 (reference)	
A-G-A	65 (12.1)	51 (16.0)	1.561 (1.001–2.433)	**0.049**
A-A-C	71 (13.2)	42 (13.2)	1.177 (0.745–1.857)	0.485
A-A-A	3 (0.6)	0 (0.0)	0.283 (0.014–5.549)	0.553
G-G-C	105 (19.6)	67 (21.1)	1.269 (0.854–1.886)	0.237
G-G-A	51 (9.5)	33 (10.4)	1.287 (0.777–2.132)	0.326
G-A-C	59 (11.0)	35 (11.0)	1.180 (0.725–1.921)	0.506
G-A-A	0 (0.0)	2 (0.6)	9.919 (0.471–208.9)	0.114
miR-218-2 rs11134527 G>A/miR-34a rs2666433 G>A/miR-130a rs731384 G>A				
A-G-G	207 (38.6)	132 (41.5)	1.000 (reference)	
A-G-A	40 (7.5)	13 (4.1)	0.515 (0.265–0.998)	**0.046**
A-A-G	70 (13.1)	32 (10.1)	0.724 (0.452–1.16)	0.178
A-A-A	4 (0.7)	8 (2.5)	3.167 (0.935–10.73)	0.071
G-G-G	140 (26.1)	78 (24.5)	0.882 (0.62–1.255)	0.485
G-G-A	15 (2.8)	20 (6.3)	2.111 (1.044–4.269)	**0.034**
G-A-G	49 (9.1)	39 (12.3)	1.260 (0.785–2.024)	0.338
G-A-A	11 (2.1)	0 (0.0)	0.069 (0.004–1.177)	**0.008**

Note: 95% CI, 95% confidence interval; OR, odds ratio; N/A, not applicable.

**Table 5 ijms-24-16794-t005:** Combined genotype analysis for miRNA polymorphisms in RIF patients and controls.

Genotype Combinations	Controls (n = 268)	RIF Patients (n = 161)	AOR (95% CI)	*p*
miR-34a rs6577555 C>A/miR-130a rs731384 G>A				
CC/GG	121 (45.1)	69 (42.9)	1.000 (reference)	
CC/GA	38 (14.2)	19 (11.8)	0.865 (0.462–1.619)	0.651
CC/AA	2 (0.7)	2 (1.2)	1.745 (0.24–2.692)	0.583
CA/GG	73 (27.2)	41 (25.5)	0.975 (0.6–1.585)	0.918
CA/GA	22 (8.2)	15 (9.3)	1.189 (0.578–2.445)	0.637
CA/AA	0 (0.0)	0 (0.0)	N/A	N/A
AA/GG	8 (3.0)	13 (8.1)	2.881 (1.132–7.332)	**0.026**
AA/GA	3 (1.1)	1 (0.6)	0.627 (0.063–6.274)	0.691
AA/AA	1 (0.4)	1 (0.6)	1.825 (0.112–9.764)	0.673

Note: 95% CI, 95% confidence interval; AOR, adjusted odds ratio; N/A, not applicable.

**Table 6 ijms-24-16794-t006:** Clinical variables in RIF patients stratified by miRNA polymorphisms status by ANOVA and Kruskal–Wallis test.

Genotype	TSH (mU/L)	FSH (mU/L)	LH (mU/L)	Prolactin (ng/mL)	CD3 (Pan T)	CD4 (Helper T)	CD8 (Suppressor T)	CD19 (B-Cell)
	mean ± SD	mean ± SD	mean ± SD	mean ± SD	mean ± SD	mean ± SD	mean ± SD	mean ± SD
miR-218-2 rs11134527 G>A								
AA	2.52 ± 1.79	8.92 ± 5.47	4.88 ± 2.67	16.09 ± 8.30	65.66 ± 8.94	36.15 ± 9.80	26.08 ± 7.54	12.53 ± 4.30
AG	2.08 ± 1.23	9.10 ± 3.56	4.73 ± 2.31	12.72 ± 5.05	67.03 ± 12.48	34.17 ± 9.21	29.88 ± 8.51	11.41 ± 5.27
GG	2.15 ± 1.39	10.59 ± 7.93	5.67 ± 2.26	12.59 ± 9.91	72.37 ± 6.70	40.43 ± 7.36	26.15 ± 5.46	11.13 ± 3.67
*p* ^a^	0.339	0.325 ^b^	0.403	0.104 ^b^	0.123	0.052	**0.047**	0.485
miR-34a rs2666433 G>A								
GG	2.38 ± 1.62	9.15 ± 3.67	4.59 ± 2.02	14.95 ± 7.37	66.64 ± 9.52	35.80 ± 9.20	27.63 ± 7.74	11.60 ± 5.48
GA	1.88 ± 0.90	9.13 ± 5.99	5.28 ± 2.71	12.27 ± 6.79	67.79 ± 12.19	35.32 ± 9.26	29.03 ± 8.51	12.21 ± 3.92
AA	4.62 ± 2.48	12.64 ± 8.18	5.79 ± 4.95	10.01 ± 4.57	71.52 ± 14.77	36.98 ± 12.57	27.48 ± 7.40	8.67 ± 2.67
*p* ^a^	**0.012 ^b^**	0.479	0.476 ^b^	0.134	0.558	0.909	0.659	0.147 ^b^
miR-34a rs6577555 C>A								
CC	2.51 ± 1.54	9.33 ± 5.31	4.87 ± 2.72	13.61 ± 6.48	68.36 ± 10.09	34.48 ± 9.59	30.02 ± 7.27	10.63 ± 3.64
CA	1.94 ± 1.24	8.90 ± 4.47	5.17 ± 2.15	12.02 ± 6.27	65.71 ± 13.12	37.13 ± 8.30	25.64 ± 8.41	13.62 ± 5.49
AA	1.44 ± 0.98	9.64 ± 4.19	4.41 ± 1.14	19.48 ± 11.12	67.64 ± 7.96	37.00 ± 10.75	27.09 ± 8.66	11.23 ± 6.01
*p* ^a^	**0.020**	0.872	0.633	**0.022**	0.512	0.344	**0.027**	**0.023 ^b^**
miR-130a rs731384 G>A								
GG	2.25 ± 1.52	9.40 ± 4.98	4.90 ± 2.48	14.04 ± 7.71	66.93 ± 11.88	36.34 ± 9.58	27.57 ± 7.95	11.81 ± 4.76
GA	2.12 ± 1.19	8.70 ± 4.88	4.68 ± 2.00	12.22 ± 4.57	69.54 ± 7.74	34.08 ± 8.47	30.18 ± 8.28	11.83 ± 4.96
AA	2.15 ± 1.25	9.02 ± 3.46	7.52 ± 4.04	13.42 ± 1.65	62.33 ± 2.52	30.00 ± 7.94	29.67 ± 7.64	7.67 ± 3.21
*p* ^a^	0.922	0.832	0.283	0.631	0.420	0.326	0.345	0.338

Note: ANOVA, analysis of variance; TSH, thyroid-stimulating hormone; FSH, follicle stimulating hormone; LH, luteinizing hormone; WBC, white blood cell; Hgb, hemoglobin; SD, standard deviation. ^a^ Calculated using ANOVA. ^b^ Calculated using the Kruskal-Wallis test.

**Table 7 ijms-24-16794-t007:** Stratified analysis of miRNA polymorphisms according to risk factors in RIF patients and controls.

Variables	miR-218-2 (rs11134527 G>A) AG+GG		miR-34a (rs2666433 G>A) GA+AA		miR-34a (rs6577555 C>A) CA+AA		miR-130a (rs731384 G>A) GA+AA	
	AOR (95% CI)	*p*	AOR (95% CI)	*p*	AOR (95% CI)	*p*	AOR (95% CI)	*p*
BMI (kg/m^2^)								
<24.2	1.141 (0.613–2.124)	0.677	1.691 (0.911–3.138)	0.096	0.814 (0.45–1.474)	0.498	1.485 (0.71–3.104)	0.293
≥24.2	2.311 (0.478–1.165)	0.297	0.42 (0.105–1.683)	0.22	2.262 (0.539–9.497)	0.265	0.369 (0.081–1.669)	0.195
PT (s)								
<11.7	1.341 (0.778–2.312)	0.29	0.83 (0.502–1.373)	0.468	**1.731 (1.037–2.892)**	**0.036**	1.089 (0.593–1.999)	0.784
≥11.7	0.688 (0.167–2.844)	0.606	2.614 (0.595–1.481)	0.203	0.802 (0.192–3.354)	0.762	N/A	N/A
aPTT (s)								
<33	1.072 (0.632–1.817)	0.797	0.73 (0.442–1.204)	0.217	1.49 (0.897–2.472)	0.123	1.27 (0.686–2.35)	0.447
≥33	1.774 (0.515–6.105)	0.364	**3.59 (1.014–12.71)**	**0.048**	1.871 (0.522–6.704)	0.336	0.865 (0.225–3.322)	0.833
PLT (10^3^/μL)								
<301	1.185 (0.733–1.918)	0.488	1.223 (0.775–1.931)	0.387	1.141 (0.721–1.806)	0.573	0.935 (0.546–1.601)	0.806
≥301	1.801 (0.542–5.987)	0.337	0.945 (0.323–2.767)	0.917	1.844 (0.61–5.573)	0.278	0.986 (0.299–3.251)	0.982
Homocysteine (μmol/L)								
<8.7	0.798 (0.188–3.389)	0.76	1.006 (0.273–3.699)	0.993	0.674 (0.176–2.579)	0.564	3.718 (0.422–2.731)	0.237
≥8.7	0 (0–0)	0.994	0.667 (0.029–5.217)	0.799	0.548 (0.022–3.889)	0.715	0.324 (0.011–9.513)	0.513
Folate (mg/mL)								
>6.8	0.469 (0.11–2.002)	0.307	0.817 (0.248–2.686)	0.739	0.511 (0.148–1.761)	0.288	1.993 (0.365–0.886)	0.426
≤6.8	N/A	N/A	N/A	N/A	N/A	N/A	N/A	N/A
Uric acid (mg/dl)								
<5	1.448 (0.745–2.815)	0.275	1.118 (0.602–2.078)	0.724	2.296 (1.211–4.354)	**0.011**	1.902 (0.853–4.242)	0.116
≥5	1.65 (0.348–7.822)	0.528	0.876 (0.203–3.776)	0.859	1.101 (0.249–4.87)	0.899	0.648 (0.145–2.9)	0.57
BUN (mg/dl)								
<12.6	1.789 (0.972–3.295)	0.062	1.163 (0.677–1.999)	0.584	1.248 (0.711–2.192)	0.44	1.131 (0.57–2.246)	0.725
≥12.6	0.322 (0.077–1.348)	0.121	0.359 (0.081–1.592)	0.178	1.739 (0.401–7.546)	0.46	0.507 (0.117–2.197)	0.364
Creatinine (mg/dl)								
<0.9	1.445 (0.802–2.605)	0.221	0.959 (0.556–1.652)	0.879	1.545 (0.879–2.716)	0.13	1.589 (0.833–3.031)	0.16
≥0.9	0.435 (0.109–1.738)	0.239	1.529 (0.424–5.518)	0.517	0.324 (0.089–1.186)	0.089	0.596 (0.098–3.632)	0.575
E2 (Basal) (pg/mL)								
<48.5	1.298 (0.706–2.384)	0.402	1.326 (0.74–2.378)	0.344	1.055 (0.59–1.886)	0.857	0.772 (0.41–1.455)	0.424
≥48.5	1.907 (0.333–0.922)	0.469	1.512 (0.241–9.467)	0.659	0.513 (0.095–2.758)	0.436	0.37 (0.063–2.171)	0.271
TSH (mU/L)								
>0.849	0.917 (0.468–1.798)	0.802	1.073 (0.582–1.98)	0.821	1.071 (0.563–2.037)	0.834	1.346 (0.636–2.847)	0.438
≤0.849	1.229 (0.229–6.608)	0.81	0.264 (0.046–1.499)	0.133	**20.243 (2.836–144.5)**	**0.003**	0.749 (0.112–5.025)	0.766
FSH (mU/L)								
<11.41	1.373 (0.736–2.56)	0.319	1.548 (0.846–2.831)	0.157	1.013 (0.56–1.833)	0.965	0.818 (0.429–1.561)	0.542
≥11.41	0.496 (0.082–2.996)	0.445	0.502 (0.102–2.479)	0.397	1.613 (0.335–7.765)	0.551	0.393 (0.06–2.556)	0.328
LH (mU/L)								
<6.48	1.176 (0.626–2.21)	0.615	1.095 (0.593–2.021)	0.772	0.978 (0.537–1.784)	0.943	0.713 (0.366–1.389)	0.32
≥6.48	2.005 (0.335–2.01)	0.446	1.412 (0.251–7.928)	0.695	0.588 (0.103–3.359)	0.551	0.803 (0.109–5.932)	0.829
WBC (10^3^/μL)								
<9.8	1.201 (0.726–1.989)	0.476	1.208 (0.75–1.946)	0.436	0.997 (0.613–1.621)	0.989	0.922 (0.53–1.605)	0.775
≥9.8	0.822 (0.242–2.796)	0.754	1.321 (0.421–4.148)	0.633	2.951 (0.906–9.615)	0.073	2.364 (0.685–8.158)	0.174
Hgb (g/dl)								
>11.2	1.192 (0.722–1.968)	0.492	1.183 (0.731–1.912)	0.494	1.154 (0.712–1.87)	0.56	0.827 (0.474–1.445)	0.505
≤11.2	1 (0.284–3.527)	1	1.064 (0.333–3.397)	0.917	1.413 (0.441–4.526)	0.56	**4.787 (1.212–8.906)**	**0.026**

Note: BMI, body mass index; PT, prothrombin time; aPTT, activated partial thromboplastin time; PLT, platelet count; BUN, blood urea nitrogen; E2, estradiol; TSH, thyroid-stimulating hormone; FSH, follicle stimulating hormone; LH, luteinizing hormone; WBC, white blood cell; Hgb, hemoglobin; 95% CI, 95% confidence interval; AOR, adjusted odds ratio; N/A, not applicable; AOR was adjusted by age. Folate, TSH and Hgb were lower 15% cut-off each level in RIF patients and controls; BMI, PT, aPTT, PLT, Homocysteine, Uric acid, BUN, Creatinine, E2, FSH, LH, and WBC were upper 15% cut-off each level in RIF patients and controls.

## Data Availability

The data presented in this study can be made available upon request from the corresponding author.

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
