# Peer review of "Genetic Correlation of miRNA Polymorphisms and STAT3 Signaling Pathway with Recurrent Implantation Failure in the Korean Population"

_ijms, 2023, doi:10.3390/ijms242316794_

Round 1

Reviewer 1 Report

Comments and Suggestions for Authors

The main objective of this study is to investigate -in Korean population- the potential association between SNPs in specific microRNAs, and the risk of recurrent implantation failure (RIF), shedding light on the genetic factors contributing to RIF. The specific microRNAs chosen for analysis are gene expression modulators of STAT3, whose activation is known to be essential for embryo implantation. After analyzing the paper it appears that the AA genotype of one of those microRNAs contributes to RIF susceptibility by triggering inflammation through the modulation of STAT3 gene expression (decreasing it), and that it also correlates with lower levels of TSH. The main contribution of research paper is that it established microRNAs as biomarkers for RIF, and as possible therapeutic targets in the future.  Strength of this article is the abudancy of statistical analysis and the precise knowledge of gene expression interactions that allow understanding of statistical correlations. Results and discussion are presented before maaterials and methods. This is unusual but it allows following a logical line of thought to understand the research.

All sections of the article serve well their purpose. Title adequately describes the content of the article and the abstract is well structured. Introduction is ordered from background to current knowledge and presents objective of the research. Results present extensive tables but the bolded values and the text explanations allow full understanding of the relevant results. Discussion is adecuate and links results with background, reflected concisely and clearly in the final conclusion of the article. Materials and methods provide sufficient detail for reproductibility. 

However, a minor remarks that needs to be addressed is that in line 327 the description of the number of subjects included in the RIF and control group do not match those described in the abstract. Also, the reference citation on line 436 doesn't match the citation format of the rest of the article. 

Other than that, my asessment is that the article is suitable for publication

Reviewer 2 Report

Comments and Suggestions for Authors

The study titled "Genetic correlation of miRNA polymorphisms and STAT3 signaling pathway with recurrent implantation failure in the Korean population" is a very well-orchestrated study and the results and analysis were also neatly executed. Overall, the outcome of the study is also very useful to the scientific community involved in understanding the reasons for recurrent implantation failures of IVF.

According to my knowledge, especially with respect to the Asian
    population, this kind of clinical samples involving studies are very
    sparse that too involving the reproductive failure cases. Therefore,
    I would like to appreciate the study team for taking up challenging
    research, which is pertaining to the genetic analyses of the
    subjects, especially those who are experiencing frequent failure of
    embryo implantation. However, I wonder how a study team recruited
    these study participants nearly 10 years after obtaining the ethical
    approval for this type of study. To my knowledge, the ethical
    approval for any study would be given for a stipulated period of
    time, especially maximum of 3 years. Kindly clarify in this regard.

    I have to appreciate this research because in general, the role of
    SNPs in coding genes have well been documented so far for several
    diseases. However, it is interesting to note that this study tried a
    novel aspect of identifying the regulatory SNPs in miRNAs that are
    involved in controlling the key inflammatory genes like STAT3 and
    IL6 for their role in successful embryo implantation in couples
    facing reproductive challenges.

    Regarding methodology part, the SNPs analysis by Genotyping can be
    more elaborated including the details of step-by-step procedure so
    that the other studies can follow and adapt this research protocol
    in future. No doubt, the authors have used standard kits perform
    these analyses, however, it would be great if they provide how much
    sample has been collected from the participants and how much sample
    they have used for the extraction of DNA, how the quantity and
    quality of DNA has been analysed, quantity of DNA used for the SNP
    genotyping, etc. key aspects would helpful for the readers of this
    journal.

    As per the conclusion part concerned, I agree that the authors have
    concluded the study properly and also mentioned the drawbacks of
    this study too. So, I feel it is appropriate to the research
    question that they have proposed.

I appreciate the entire research team and recommend this manuscript for publication in this journal.
